# Inverting Deep Generative models,
# One layer at a time

Qi Lei[†], Ajil Jalal[†], Inderjit S. Dhillon[†‡], and Alexandros G. Dimakis[†]

[†] UT Austin  [‡] Amazon
{leiqi@oden., ajiljalal@, inderjit@cs.,
dimakis@austin.}utexas.edu

## Abstract

We study the problem of inverting a deep generative model with ReLU activations. Inversion corresponds to finding a latent code vector that explains observed measurements as much as possible. In most prior works this is performed by attempting to solve a non-convex optimization problem involving the generator. In this paper we obtain several novel theoretical results for the inversion problem.

We show that for the realizable case, single layer inversion can be performed exactly in polynomial time, by solving a linear program. Further, we show that for multiple layers, inversion is NP-hard and the pre-image set can be non-convex.

For generative models of arbitrary depth, we show that exact recovery is possible in polynomial time with high probability, if the layers are expanding and the weights are randomly selected. Very recent work analyzed the same problem for gradient descent inversion. Their analysis requires significantly higher expansion (logarithmic in the latent dimension) while our proposed algorithm can provably reconstruct even with constant factor expansion. We also provide provable error bounds for different norms for reconstructing noisy observations. Our empirical validation demonstrates that we obtain better reconstructions when the latent dimension is large.

## 1 Introduction

Modern deep generative models are demonstrating excellent performance as signal priors, frequently outperforming the previous state of the art for various inverse problems including denoising, inpainting, reconstruction from Gaussian projections and phase retrieval (see e.g. [4, 6, 10, 5, 11, 25] and references therein). Consequently, there is substantial work on improving compressed sensing with generative adversarial network (GANs) [9, 17, 13, 18, 20]. Similar ideas have been recently applied also for sparse PCA with a generative prior [2].

A central problem that appears when trying to solve inverse problems using deep generative models is *inverting a generator* [4, 12, 24]. We are interested in deep generative models, parameterized as feed-forward neural networks with ReLU/LeakyReLU activations. For a generator $G(z)$ that maps low-dimensional vectors in $\mathbb{R}^k$ to high dimensional vectors (e.g. images) in $\mathbb{R}^n$, we want to reconstruct the latent code $z^*$ if we can observe $x = G(z^*)$ (realizable case) or a noisy version $x = G(z^*) + e$ where $e$ denotes some measurement noise. We are therefore interested in the optimization problem

$$\arg\min_{z} \|x - G(z)\|_p, \tag{1}$$

for some $p$ norm. With this procedure, we learn a concise image representation of a given image $x \in \mathbb{R}^n$ as $z \in \mathbb{R}^k, k \ll n$. This applies to image compressions and denoising tasks as studied in

[14, 13]. Meanwhile, this problem is a starting point for general linear inverse problems:

$$\arg\min_{\boldsymbol{z}} \|\boldsymbol{x} - AG(\boldsymbol{z})\|_p, \tag{2}$$

since several recent works leverage inversion as a key step in solving more general inverse problems, see e.g. [24, 22]. Specifically, Shah et al. [24] provide theoretical guarantees on obtaining the optimal solution for (2) with projected gradient descent, provided one could solve (1) exactly. This work provides a provable algorithm to perform this projection step under some assumptions.

Previous work focuses on the $\ell_2$ norm that works slowly with gradient descent [4, 15]. In this work, we focus on direct solvers and error bound analysis for $\ell_\infty$ and $\ell_1$ norm instead.[1] Note that this is a non-convex optimization problem even for a single-layer network with ReLU activations. Therefore gradient descent may get stuck at local minimima or require a long time to converge. For example, for MNIST, compressing a single image by optimizing (1) takes on average several minutes and may need multiple restarts.

**Our Contributions:** For the realizable case we show that for a single layer solving (1) is equivalent to solving a linear program. For networks more than one layer, however, we show it is NP-hard to simply determine whether exact recovery exists. For a two-layer network we show that the pre-image in the latent space can be a non-convex set.

For realizable inputs and arbitrary depth we show that inversion is possible in polynomial time if the network layers have sufficient expansion and the weights are randomly selected. A similar result was established very recently for gradient descent [15]. We instead propose inversion by layer-wise Gaussian elimination. Our result holds even if each layer is expanding by a constant factor while [15] requires a logarithmic multiplicative expansion in each layer.

For noisy inputs and arbitrary depth we propose two algorithms that rely on iteratively solving linear programs to reconstruct each layer. We establish provable error bounds on the reconstruction error when the weights are random and have constant expansion. We also show empirically that our method matches and sometimes outperforms gradient descent for inversion, especially when the latent dimension becomes larger.

## 2 Setup

We consider deep generative models $G : \mathbb{R}^k \to \mathbb{R}^n$ with the latent dimension $k$ being smaller than the signal dimension $n$, parameterized by a $d$-layer feed-forward network of the form

$$G(\boldsymbol{z}) = \phi_d(\phi_{d-1}(\cdots \phi_2(\phi_1(\boldsymbol{z}))\cdots)), \tag{3}$$

where each layer $\phi_i(\boldsymbol{a})$ is defined as a composition of activations and linear maps: $\text{ReLU}(W_i\boldsymbol{a} + \boldsymbol{b}_i)$. We focus on the ReLU activations $\text{ReLU}(\boldsymbol{a}) = \max\{\boldsymbol{a}, \boldsymbol{0}\}$ applied coordinate-wise, and we will also consider the activation as $\text{LeakyReLU}(\boldsymbol{a}) = \text{ReLU}(\boldsymbol{a}) + c\text{ReLU}(-\boldsymbol{a})$, where the scaling factor $c \in (0, 1)$ is typically $0.1$.[2] $W_i \in \mathbb{R}^{n_i \times n_{i-1}}$ are the weights of the network, and $\boldsymbol{b}_i \in \mathbb{R}^{n_i}$ are the bias terms. Therefore, $n_0 = k$ and $n_d = n$ indicate the dimensionality of the input and output of the generator $G$. We use $\boldsymbol{z}_i$ to denote the output of the $i$-th layer. Note that one can absorb the bias term $\boldsymbol{b}_i, i = 1, 2, \cdots d$ into $W_i$ by adding one more dimension with a constant input. Therefore, without loss of generality, we sometimes omit $\boldsymbol{b}_i$ when writing the equation, unless we explicitly needed it.

We use bold lower-case symbols for vectors, e.g. $\boldsymbol{x}$, and $x_i$ for its coordinates. We use upper-case symbols for denote matrices, e.g. $W$, where $\boldsymbol{w}_i$ is its $i$-th row vector. For a indexed set $I$, $W_{I,:}$ represents the submatrix of $W$ consisting of each $i$-th row of $W$ for any $i \in I$.

The central challenge is to determine the signs for the intermediate variables of the hidden layers. We refer to these sign patterns as "ReLU configurations" throughout the paper, indicating which neurons are 'on' and which are 'off'.

## 3 Invertibility for ReLU Realizable Networks

In this section we study the realizable case, i.e., when we are given an observation vector $\boldsymbol{x}$ for which there exists $\boldsymbol{z}^*$ such that $\boldsymbol{x} = G(\boldsymbol{z}^*)$. In particular, we show that the problem is NP-hard for ReLU

activations in general, but could be solved in polynomial time with some mild assumptions with high probability. We present our theoretical findings first and all proofs of the paper are presented later in the Appendix.

**Inverting a Single Layer.** We start with the simplest one-layer case to find if $\min_{\boldsymbol{z}} \|\boldsymbol{x} - G(\boldsymbol{z})\|_p = 0$, for any $p$-norm. Since the problem is non-convex, further assumptions of $W$ are required [15] for gradient descent to work. When the problem is realizable, however, to find feasible $\boldsymbol{z}$ such that $\boldsymbol{x} = \phi(\boldsymbol{z}) \equiv \text{ReLU}(W\boldsymbol{z} + \boldsymbol{b})$, one could invert the function by solving a linear programming:

$$\boldsymbol{w}_i^\top \boldsymbol{z} + b_i = x_i, \quad \forall i \text{ s.t. } x_i > 0$$
$$\boldsymbol{w}_i^\top \boldsymbol{z} + b_i \leq 0, \quad \forall i \text{ s.t. } x_i = 0 \quad\quad (4)$$

Its solution set is convex and forms a polytope, but possibly includes uncountable feasible points. Therefore, it becomes unclear how to continue the process of layer-wise inversion unless further assumptions are made. To demonstrate the challenges to generalize the result to deeper nets, we show that the solution set becomes non-convex, and to determine whether there exists any solution is NP-complete.

**Challenges to Invert a Two or More Layered ReLU Network.**

We would like to study the complexity of inverting deep ReLU networks in general. We do this by constructing a 4-layer network and prove the following statement:

**Theorem 1** (NP-hardness to Recover ReLU Networks with Real Domain). *Given a four-layered ReLU neural network $G(\boldsymbol{x}) : \mathbb{R}^k \to \mathbb{R}^2$ where weights are all fixed, and an observation vector $x \in \mathbb{R}^2$, the problem to determine whether there exists $\boldsymbol{z} \in \mathbb{R}^k$ such that $G(\boldsymbol{z}) = \boldsymbol{x}$ is NP-complete.*

The conclusion holds naturally for generative models with deeper architecture. We defer the proof to the Appendix, which is constructive and shows the 3SAT problem is reducible to the above four-layer network recovery problem. Meanwhile, when the ReLU configuration for each layer is given, the recovery problem becomes to solve a simple linear system. Therefore the problem lies in NP, and together we have NP-completeness.

Meanwhile, although the pre-image for a single layer is a polytope thus convex, it doesn't continue to hold for more than one layers, see Example 1. Fortunately, we present next that some moderate conditions guarantee a polynomial time solution with high probability.

**Inverting Expansive Random Network in Polynomial Time.**

**Assumption 1.** *For a weight matrix $W \in \mathbb{R}^{n \times k}$, we assume 1) its entries are sampled i.i.d Gaussian, and 2) the weight matrix is tall: $n = c_0 k$ for some constant $c_0 \geq 2.1$.*

In the previous section, we indicate that the per layer inversion can be achieved through linear programming (4). With Assumption 1 we will be able to prove that the solution is unique with high probability, and thus Theorem 2 holds for ReLU networks with arbitrary depth.

**Theorem 2.** *Let $G \in \mathbb{R}^k \to \mathbb{R}^n$ be a generative model from a $d$-layer neural network using ReLU activations. If for each layer, the weight matrix $W_i$ satisfies Assumption 1, then for any prior $\boldsymbol{z}^* \in \mathbb{R}^k$ and observation $\boldsymbol{x} = G(\boldsymbol{z}^*)$, with probability $1 - e^{-\Omega(k)}$, $\boldsymbol{z}^*$ could be achieved from $\boldsymbol{x}$ by solving layer-wise linear equations. Namely, a random, expansive and realizable generative model could be inverted in polynomial time with high probability.*

In our proof, we show that with high probability the observation $\boldsymbol{x} \in \mathbb{R}^n$ has at least $k$ non-zero entries, which forms $k$ equalities and the coefficient matrix is invertible with probability 1. Therefore the time complexity of exact recovery is no worse than $\sum_{i=0}^{d-1} n_i^{2.376}$ [7] since the recovery simply requires solving $d$ linear equations with dimension $n_{i-1}, i \in [d]$.

**Inverting LeakyReLU Network:** On the other hand, inversion of LeakyReLU layers are significantly easier for the realizable case. Unlike ReLU, LeakyReLU is a bijective map, i.e., each observation corresponds to a unique preimage:

$$\text{LeakyReLU}^{-1}(x) = \begin{cases} x & \text{if } x \geq 0 \\ 1/cx & \text{otherwise.} \end{cases} \quad\quad (5)$$

Therefore, as long as each $W_i \in \mathbb{R}^{n_i \times n_{i-1}}$ is of rank $n_{i-1}$, each layer map $\phi_i$ is also bijective and could be computed by the inverse of LeakyReLU (5) and linear regression.

# 4 Invertibility for Noisy ReLU Networks

Besides the realizable case, the study of noise tolerance is essential for many real applications. In this section, we thus consider the noisy setting with observation $\boldsymbol{x} = G(\boldsymbol{z}^*) + \boldsymbol{e}$, and investigate the approximate recovery for $\boldsymbol{z}^*$ by relaxing some equalities in (4). We also analyze the problem with both $\ell_\infty$ and $\ell_1$ error bound, in favor of different types of random noise distribution. In this section, all generators are without the bias term.

## 4.1 $\ell_\infty$ Norm Error Bound

Again we start with a single layer, i.e. we observe $\boldsymbol{x} = \phi(\boldsymbol{z}^*) + \boldsymbol{e} = \text{ReLU}(W\boldsymbol{z}^*) + \boldsymbol{e}$. Depending on the distribution over the measurement noise $\boldsymbol{e}$, different norm in the objective $\|G(\boldsymbol{z}) - \boldsymbol{x}\|$ should be used, with corresponding error bound analysis. We first look at the case where the entries of $\boldsymbol{e}$ are uniformly bounded and the approximation of $\arg\min_{\boldsymbol{z}} \|\phi(\boldsymbol{z}) - \boldsymbol{x}\|_\infty$.

Note that for an error $\|\boldsymbol{e}\|_\infty \leq \epsilon$, the true prior $\boldsymbol{z}^*$ that produces the observation $\boldsymbol{x} = \phi(\boldsymbol{z}^*) + \boldsymbol{e}$ falls into the following constraints:

$$
\begin{aligned}
x_j - \epsilon \leq \boldsymbol{w}_j^\top \boldsymbol{z} \leq x_j + \epsilon \quad &\text{if } x_j > \epsilon, j \in [n] \\
\boldsymbol{w}_j^\top \boldsymbol{z} \leq x_j + \epsilon \quad &\text{if } x_j \leq \epsilon, j \in [n],
\end{aligned}
\tag{6}
$$

which is also equivalent to the set $\{\boldsymbol{z} | \|\phi(\boldsymbol{z}) - \boldsymbol{x}\|_\infty \leq \epsilon\}$. Therefore a natural way to approximate the prior is to use linear programming to solve the above constraints.

If $\epsilon$ is known, inversion is straightforward from constraints (6). However, suppose we don't want to use a loose guess, we could start from a small estimation and gradually increase the tolerance until feasibility is achieved. A layer-wise inversion is formally presented in Algorithm 1[3].

A key assumption that possibly conveys the error bound from the output to the solution is the following assumption:

**Assumption 2** (Submatrix Extends $\ell_\infty$ Norm). *For the weight matrix $W \in \mathbb{R}^{n \times k}$, there exists an integer $m > k$ and a constant $c_\infty$, such that for any $I \subset [n] := \{1, 2, \cdots n\}, |I| \geq m$, $W_{I,:}$ satisfies*

$$\|W_{I,:}\boldsymbol{x}\|_\infty \geq c_\infty \|\boldsymbol{x}\|_\infty,$$

*with high probability $1 - \exp(-\Omega(k))$ for any $\boldsymbol{x}$, and $c_\infty$ is a constant. Recall that $W_{I,:}$ is the sub-rows of $W$ confined to $I$.*

With this assumption, we are able to show the following theorem that bounds the recovery error.

**Theorem 3.** *Let $\boldsymbol{x} = G(\boldsymbol{z}^*) + \boldsymbol{e}$ be a noisy observation produced by the generator $G$, a $d$-layer ReLU network mapping from $\mathbb{R}^k \to \mathbb{R}^n$. Let each weight matrix $W_i \in \mathbb{R}^{n_{i-1} \times n_i}$ satisfies Assumption 2 with the integer $m_i > n_{i-1}$ and constant $c_\infty$. Let the error $\boldsymbol{e}$ satisfies $\|\boldsymbol{e}\|_\infty \leq \epsilon$, and for each $\boldsymbol{z}_i = \phi_i(\phi_{i-1}(\cdots \phi(\boldsymbol{z}^*) \cdots))$, at least $m_i$ coordinates are larger than $2(2/c_\infty)^{d-i}\epsilon$. Then by recursively applying Algorithm 1 backwards, it produces a $\boldsymbol{z}$ that satisfies $\|\boldsymbol{z} - \boldsymbol{z}^*\|_\infty \leq (2/c_\infty)^d \epsilon$ with high probability.*

We argue that the assumptions required are satisfied by random weight matrices sampled from an i.i.d Gaussian distribution, and present the following corollary.

**Corollary 1.** *Let $\boldsymbol{x} = G(\boldsymbol{z}^*) + \boldsymbol{e}$ be a noisy observation produced by the generator $G$, a $d$-layer ReLU network mapping from $\mathbb{R}^k \to \mathbb{R}^n$. Let each weight matrix $W_i \in \mathbb{R}^{n_{i-1} \times n_i}$ ($n_i \geq 5n_{i-1}, \forall i$) be sampled from i.i.d Gaussian distribution $\sim \mathcal{N}(0, 1)$, then $W_i$ satisfies Assumption 2 for a universal constant constant $c_2 \in (0, 2]$. Let the error $\boldsymbol{e}$ satisfy $\|\boldsymbol{e}\|_\infty = \epsilon$, where $\epsilon < \frac{c_2^d}{2^{d+4}}\|\boldsymbol{z}^*\|_2\sqrt{k}$. By recursively applying Algorithm 1, it produces $\boldsymbol{z}$ that satisfies $\|\boldsymbol{z} - \boldsymbol{z}^*\|_\infty \leq \frac{2^d \epsilon}{c_2^d}$ with high probability.*

**Remark 1.** *For LeakyReLU, we could do at least as good as ReLU, since we could simply view all negative coordinates as inactive coordinates of ReLU, and each observation will produce a loose bound. On the other hand, if there are significant number of negative entries, we can also change the*

*linear programming constraints of Algorithm 1 as follows:*

$$\arg\min_{\boldsymbol{z},\delta} \ \delta, \ s.t. \ \begin{cases} x_j - \delta \leq \boldsymbol{w}_j^\top \boldsymbol{z} \leq x_j + \delta & \text{if } x_j > \epsilon \\ 1/c(x_j - \delta) \leq \boldsymbol{w}_j^\top \boldsymbol{z} \leq x_j + \delta & \text{if } -\epsilon < x_j \leq \epsilon \\ x_j - \delta \leq c\boldsymbol{w}_j^\top \boldsymbol{z} \leq x_j + \delta & \text{if } x_j \leq -\epsilon \\ \delta \leq \epsilon. \end{cases}$$

## 4.2 $\ell_1$ Norm Error Bound

In this section we develop a generative model inversion framework using the $\ell_1$ norm. We introduce Algorithm 2 that tolerates error in different level for each output coordinate and intends to minimize the $\ell_1$ norm error bound.

---

**Algorithm 1** Linear programming to invert a single layer with $\ell_\infty$ error bound ($\ell_\infty$ LP)

---

**Input:** Observation $\boldsymbol{x} \in \mathbb{R}^n$, weight matrix $W = [\boldsymbol{w}_1 | \boldsymbol{w}_2 | \cdots | \boldsymbol{w}_n]^\top$, initial error bound guess $\epsilon > 0$, scaling factor $\alpha > 1$.
**repeat**
    Find $\arg\min_{\boldsymbol{z},\delta} \ \delta$, s.t.

$$\begin{cases} x_j - \delta \leq \boldsymbol{w}_j^\top \boldsymbol{z} \leq x_j + \delta & \text{if } x_j > \epsilon \\ \boldsymbol{w}_j^\top \boldsymbol{z} \leq x_j + \delta & \text{if } x_j \leq \epsilon \\ \delta \leq \epsilon \end{cases}$$

    $\epsilon \leftarrow \epsilon\alpha$
**until** feasible solution $\boldsymbol{z}$ is found
**Output:** $\boldsymbol{z}$

---

**Algorithm 2** Linear programming to invert a single layer with $\ell_1$ error bound ($\ell_1$ LP)

---

**Input:** Observation $\boldsymbol{x} \in \mathbb{R}^n$, weight matrix $W = [\boldsymbol{w}_1 | \boldsymbol{w}_2 | \cdots | \boldsymbol{w}_n]^\top$, initial error bound guess $\epsilon > 0$, scaling factor $\alpha > 1$.
**for** $t = 1, 2, \cdots$ **do**
    $\boldsymbol{z}^{(t)}, \boldsymbol{e}^{(t)} \leftarrow \arg\min_{\boldsymbol{z},\boldsymbol{e}} \ \sum_i e_i$, s.t.

$$\begin{cases} x_j - e_j \leq \boldsymbol{w}_j^\top \boldsymbol{z} \leq x_j + e_j & \text{if } x_j > \epsilon \\ \boldsymbol{w}_j^\top \boldsymbol{z} \leq x_j + e_j & \text{if } x_j \leq \epsilon \\ e_j \geq 0 & \forall j \in [n] \end{cases}$$

    $\epsilon \leftarrow \epsilon\alpha$
    **if** $\|\phi(\boldsymbol{z}^{(t)}) - \boldsymbol{x}\|_1 \geq \|\phi(\boldsymbol{z}^{(t-1)}) - \boldsymbol{x}\|_1$ **then**
        **return** $\boldsymbol{z}^{(t-1)}$
    **end if**
**end for**

---

Different from Algorithm 1, the deviating error allowed on each observation is no longer uniform and the new algorithm is actually optimizing over the $\ell_1$ error. Similar to the error bound analysis with $\ell_\infty$ norm we are able to get a tight approximation guarantee under some mild assumption related to Restricted Isometry Property for $\ell_1$ norm:

**Assumption 3** (Submatrix Extends $\ell_1$ Norm). *For a weight matrix $W \in \mathbb{R}^{n \times k}$, there exists an integer $m > k$ and a constant $c_1$, such that for any $I \subset [n], |I| \geq m$, $W_{I,:}$ satisfies*

$$\|W_{I,:}\boldsymbol{x}\|_1 \geq c_1\|\boldsymbol{x}\|_1, \tag{7}$$

*with high probability $1 - \exp(-\Omega(k))$ for any $\boldsymbol{x}$.*

This assumption is a special case of the lower bound of the well-studied Restricted Isometry Property, for $\ell_1$-norm and sparsity $k$, i.e., $(k, \infty)$-RIP-1. Similar to the $\ell_\infty$ analysis, we are able to get recovery guarantees for generators with arbitary depth.

**Theorem 4.** *Let $\boldsymbol{x} = G(\boldsymbol{z}^*) + \boldsymbol{e}$ be a noisy observation produced by the generator $G$, a $d$-layer ReLU network mapping from $\mathbb{R}^k \to \mathbb{R}^n$. Let each weight matrix $W_i \in \mathbb{R}^{n_{i-1} \times n_i}$ satisfy Assumption 3 with the integer $m_i > n_{i-1}$ and constant $c_1$. Let the error $\boldsymbol{e}$ satisfy $\|\boldsymbol{e}\|_1 \leq \epsilon$, and for each $\boldsymbol{z}_i = \phi_i(\phi_{i-1}(\cdots \phi(\boldsymbol{z}^*)\cdots))$, at least $m_i$ coordinates are larger than $\frac{2^{d+1-i}\epsilon}{c_1^{d-i}}$. Then by recursively applying Algorithm 2, it produces a $\boldsymbol{z}$ that satisfies $\|\boldsymbol{z} - \boldsymbol{z}^*\|_1 \leq \frac{2^d\epsilon}{c_1^d}$ with high probability.*

There is a significant volume of prior work on the RIP-1 condition. For instance, studies in [3] showed that a (scaled) random sparse binary matrix with $m = O(s\log(k/s)/\epsilon^2)$ rows is $(s, 1 + \epsilon)$-RIP-1 with high probability. In our case $s = k$ and $\epsilon$ could be arbitrarily large, therefore again we only require the expansion factor to be constant. Similar results with different weight matrices are also shown in [19, 16, 1].

### 4.3 Relaxation on the ReLU Configuration Estimation

Our previous methods critically depend on the correct estimation of the ReLU configurations. In both Algorithm 1 and 2, we require the ground truth of all intermediate layer outputs to have many coordinates with large magnitude so that they can be distinguished from noise. An incorrect estimate

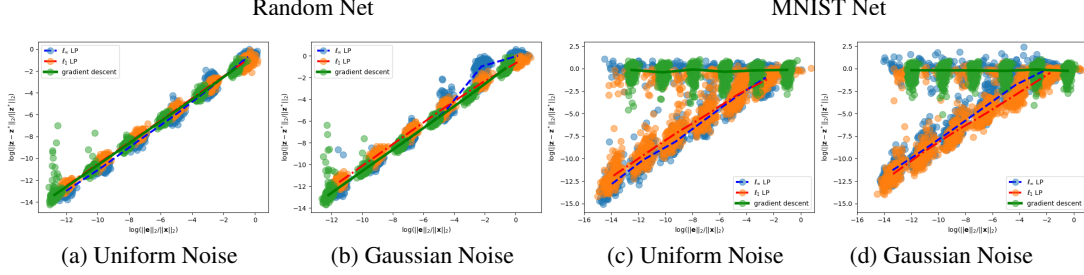

|  | Random Net |  | MNIST Net |  |
|---|---|---|---|---|
| (a) Uniform Noise | (b) Gaussian Noise | (c) Uniform Noise | (d) Gaussian Noise | |

Figure 1: Comparison of our proposed methods ($\ell_\infty$ LP and $\ell_1$ LP) versus gradient descent. On the horizontal axis we plot the relative noise level while on the vertical axis the relative recovery error. In experiments (a)(b) the network is randomly generated and fully connected, with 20 input neurons, 100 hidden neurons and 500 output neurons. This corresponds to an expansion factor of 5. Each dot represents a recovery experiment (we have 200 for each noise level). Each line connects the median of the 200 runs for each noise level. As can be seen, our algorithm (Blue and Orange) has very similar performance to gradient descent, except at low noise levels where it is slightly more robust.

In experiments (c)(d) the network is generative model for the MNIST dataset. In this case, gradient descent fails to find global minimum in almost all the cases.

from an "off" configuration to an "on" condition will possibly cause primal infeasibility when solving the LP. Increasing $\epsilon$ ameliorates this problem but also increases the recovery error.

With this intuition, a natural workaround is to perform some relaxation to tolerate incorrectly estimated signs of the observations.

$$\max_{\boldsymbol{z}} \sum_i \max\{0, x_i\} \boldsymbol{w}_i^\top \boldsymbol{z}, \ \text{s.t,} \ \boldsymbol{w}_i^\top \boldsymbol{z} \le x_i + \epsilon. \tag{8}$$

Here the ReLU configuration is no longer explicitly reflected in the constraints. Instead, we only include the upper bound for each inner product $\boldsymbol{w}_i^\top \boldsymbol{z}$, which is always valid whether the ReLU is on or off. The previous requirement for the lower bound $\boldsymbol{w}_i^\top \boldsymbol{z} \ge x_i - \epsilon$ is now relaxed and hidden in the objective part. When the value of $x_i$ is relatively large, the solver will produce a larger value of $\boldsymbol{w}_i^\top \boldsymbol{z}$ to achieve optimality. Since this value is also upper bounded by $x_i + \epsilon$, the optimal solution would be approaching to $x_i$ if possible. On the other hand, when the value of $x_i$ is close to 0, the objective dependence on $\boldsymbol{w}_i^\top \boldsymbol{z}$ is almost negligible.

Meanwhile, in the realizable case when $\exists \boldsymbol{z}^*$ such that $\text{ReLU}(W\boldsymbol{z}^*) = \boldsymbol{x}$, and $\epsilon = 0$, it is easy to show that the solution set for (8) is exactly the preimage of $\text{ReLU}(W\boldsymbol{z})$. This also trivially holds for Algorithm 1 and 2.

## 5 Experiments

In this section, we describe our experimental setup and report the performance comparisons of our algorithms with the gradient descent method [15, 12][4]. We conduct simulations in various aspects with Gaussian random weights, and a simple GAN architecture with MNIST dataset to show that our approach can work in practice for the denoising problem. We refer to our Algorithm 1 as $\ell_\infty$ LP and Algorithm 2 as $\ell_1$ LP. We focus in the main text the experiments with these two proposals and also include some more empirical findings with the relaxed version described in (8) in the Appendix.

### 5.1 Synthetic Data

We validate our algorithms on synthetic data at various noise levels and verify Theorem 3 and 4 numerically. For our methods, we choose the scaling factor $\alpha = 1.2$. With gradient descent, we use learning rate of 1 and up to 1,000 iterations or until the gradient norm is no more than $10^{-9}$.

**Model architecture:** The architecture we choose in the simulation aligns with our theoretical findings. We choose a two layer network with constant expansion factor 5: latent dimension $k = 20$, hidden neurons of size 100 and observation dimension $n = 500$. The entries in the weight matrix are independently drawn from $\mathcal{N}(0, 1/n_i)$.

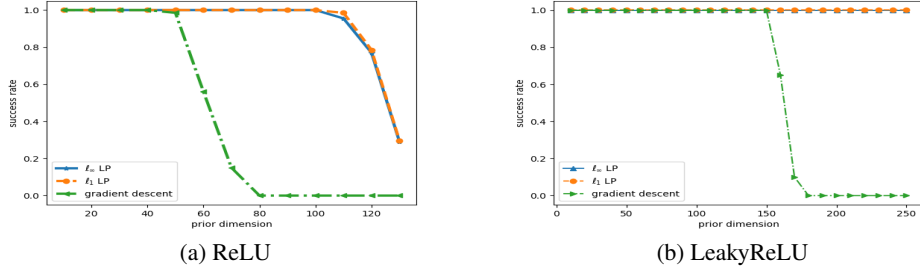

(a) ReLU  (b) LeakyReLU

Figure 2: Comparison of our method and gradient descent on the empirical success rate of recovery (200 runs on random networks) versus the number of input neurons $k$ for the noiseless problem. The architecture chosen here is a 2 layer fully connected ReLU network, with 250 hidden nodes, and 600 output neurons. Left figure is with ReLU activation and right one is with LeakyReLU. Our algorithms are significantly outpeforming gradient descent for higher latent dimensions $k$.

Observation  Ground Truth

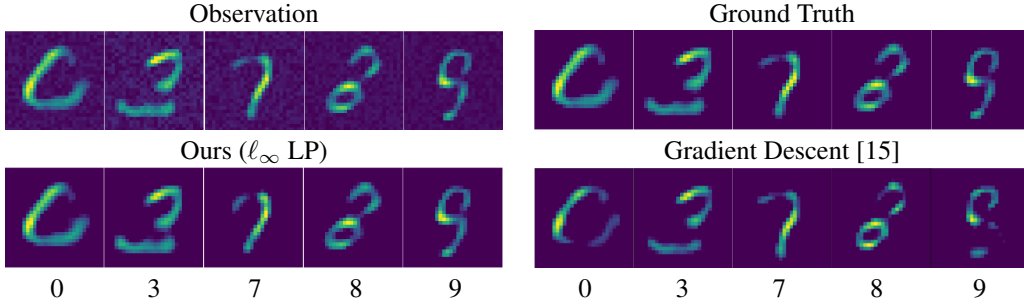

Ours ($\ell_\infty$ LP)  Gradient Descent [15]

0 3 7 8 9   0 3 7 8 9

Figure 3: Recovery comparison using our algorithm $\ell_\infty$ LP versus GD for an MNIST generative model. Notice that $\ell_\infty$ LP produces reconstructions that are clearly closer to the ground truth.

**Noise generation:** We use two kinds of random distribution to generate the noise, i.e., uniform distribution $U(-a, a)$ and Gaussian random noise $\mathcal{N}(0, a)$, in favor of the $\ell_\infty$ and $\ell_1$ error bound analysis respectively. We choose $a \in \{10^{-i} | i = 1, 2, \cdots 6\}$ for both noise types.

**Recovery with Various Observation Noise:** In Figure 1(a)(b) we plot the relative recovery error $\|z - z^*\|_2 / \|z^*\|_2$ at different noise levels. It supports our theoretical findings that with other parameters fixed, the recovery error grows almost linearly to the observation noise. Meanwhile, we observe in both cases, our methods perform similarly to gradient descent on average, while gradient descent is less robust and produces more outlier points. As expected, our $\ell_\infty$ LP performs slightly better than gradient descent when the input error is uniformly bounded; see Figure 1(a). However, with a large variance in the observation error, as seen in Figure 1(b), $\ell_\infty$ LP is not as robust as $\ell_1$ LP or gradient descent.

Additional experiments can be found in the Appendix including the performance of the LP relaxation that mimics $\ell_1$ LP but is more efficient and robust.

**Recovery with Various Input Neurons:** According to the theoretical result, one advantage of our proposals is the much smaller expansion requirement than gradient descent [12] (constant vs $\log k$ factors). Therefore we conduct the experiments to verify this point. We follow the exact setting as [15]; we fix the hidden layer and output sizes as 250 and 600 and vary the input size $k$ to measure the empirical success rate of recovery influenced by the input size.

In Figure 2 we report the empirical success rate of recovery for our proposals and gradient descent. With exact setting as in [15], a run is considered successful when $\|z^* - z\|_2 / \|z^*\|_2 \leq 10^{-3}$. We observe that when input width $k$ is small, both gradient descent and our methods grant 100% success rate. However, as the input neurons grows, gradient descent drops to complete failure when $k \geq 60$, while our algorithms continue to present 100% success rate until $k = 109$. The performance of gradient descent is slightly worse than reported in [15] since they have conducted 150 number of measurements for each run while we only considered the measurement matrix as identity matrix.

## 5.2 Experiments on Generative Model for MNIST Dataset

To verify the practical contribution of our model, we conduct experiments on a real generative network with the MNIST dataset. We set a simple fully-connected architecture with latent dimension $k = 20$,

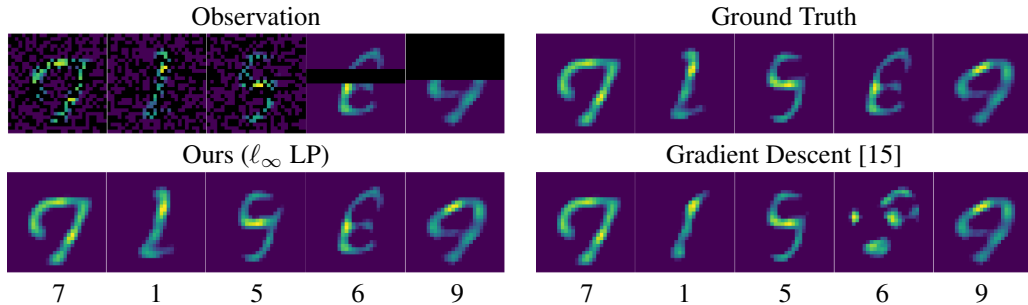

Figure 4: Recovery comparison with non-identity sensing matrix using our algorithm $\ell_\infty$ LP versus GD, for an MNIST generative model. The black region denotes unobserved pixels. Our algorithm always finds reasonable results while GD sometimes gets stuck at local minimum (See cases with number 1 and 5).

hidden neurons of size $n_1 = 60$ and output size $n = 784$. The network has a single channel. We train the network using the original Generative Adversarial Network [8]. We set $n_1$ to be small since the output usually only has around 70 to 100 non-zero pixels.

Similar to the simulation part, we compared our methods with gradient descent [12, 15]. Under this setting, we choose the learning rate to be $10^{-3}$ and number of iterations up to 10,000 (or until gradient norm is below $10^{-9}$).

We first randomly select some empirical examples to visually show performance comparison in Figure 3. In these examples, observations are perturbed with some Gaussian random noise with variance 0.3 and we use $\ell_\infty$ LP as our algorithm to invert the network. From the figures, we see that our method can almost perfectly denoise and reconstruct the input image, while gradient descent impairs the completeness of the original images to some extent.

We also compare the distribution of relative recovery error with respect to different input noise levels, as ploted in Figure 1(c)(d). From the figures, we observe that for this real network, our proposals still successfully recover the ground truth with good accuracy most of the time, while gradient descent usually gets stuck in local minimum. This explains why it produces defective image reconstructions as shown in 3.

Finally, we presented some sensing results when we mask part of the observations using PGD with our inverting procedure. As shown in Figure 4, our algorithm always show reliable recovery while gradient descent sometimes fails to output reasonable result. More experiments are presented in the Appendix.

## 6 Conclusion and Future Work

We introduced a novel algorithm to invert a generative model through linear programming, one layer at a time, given (noisy) observations of its output. We prove that for expansive and random Gaussian networks, we can exactly recover the true latent code in the noiseless setting. For noisy observations we also establish provable performance bounds. Our work is different from the closely related [15] since we require less expansion, we bound for $\ell_1$ and $\ell_\infty$ norm (as opposed to $\ell_2$), and we also only focus on inversion, i.e., without a forward operator. Our method can be used as a projection step to solve general linear inverse problems with projected gradient descent [24]. Empirically we demonstrate good performance, sometimes outperforming gradient descent when the latent vectors are high dimensional.

One message we want to convey in the paper is that it is always easier to invert to the intermediate layer than directly to the input layer. As an extreme case, we invert one layer at a time, assuming that each inversion is uniquely determined. To the best of our knowledge, all existing theoretical guarantees for inversion of deep generative models require expansion at each layer; however, models like DCGAN[21] are expansive at all layers except the output layer. In future work, we will blend our algorithms with gradient descent and propose more practical inversion algorithms.

**Acknowledgements.** This research has been supported by NSF Grants 1618689, IIS-1546452, CCF-1564000, DMS 1723052, CCF 1763702, AF 1901292 and research gifts by Google, Western Digital and NVIDIA.

## Footnotes

[1]Notice the relation between $\ell_p$ norm guarantees $\ell_p \geq \ell_q, 1 \leq p \leq q \leq \infty$. Therefore the studies on $\ell_1$ and $\ell_\infty$ is enough to bound all intermediate $\ell_p$ norms for $p \in [1, \infty)$.

[2]The inversion of LeakyReLU networks is much easier than ReLU networks and we therefore only mention it when needed.

[3]For practical use, we introduce a factor $\alpha$ to gradually increase the error estimation. In our theorem, it assumed we expicitly set $\epsilon$ to invert the $i$-th layer as the error estimation $\|\boldsymbol{e}\|_0(1/c_2)^{d-i}$.

[4]The code to reproduce our results could be found here: `https://github.com/cecilialeiqi/InvertGAN_LP`.

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
