[Supplementary Material]

## A  Methodology Details

In this section we present the detailed steps for our proposed methods.

**LeakyReLU.**

Firstly we add some remarks for LeakyReLU with tolerance of $\ell_1$ error.

**Remark 2.** *Similar to $\ell_\infty$ LP, we could extend this $\ell_1$ LP to LeakyReLU by simply adding similar constraints for the negative observations.*

$$\boldsymbol{z}^{(t)}, \boldsymbol{e}^{(t)} \leftarrow \arg\min_{\boldsymbol{z},\boldsymbol{e}} \sum_i e_i$$

$$
\begin{aligned}
s.t. \quad & x_j - e_j \leq \boldsymbol{w}_j^\top \boldsymbol{z} \leq x_j + e_j & & \text{if } x_j > \epsilon \\
& 1/c(x_j - e_j) \leq \boldsymbol{w}_j^\top \boldsymbol{z} \leq x_j + e_j & & \text{if } \epsilon \leq x_j \leq \epsilon \\
& x_j - e_j \leq c\boldsymbol{w}_j^\top \boldsymbol{z} \leq x_j + e_j & & \text{if } x_j < -\epsilon \\
& e_j \geq 0 & & \forall j \in [n].
\end{aligned}
$$

**LP Relaxation.**

We formally present the relaxed version based on (8):

---

**Algorithm 3** Relaxed Linear programming to invert a single layer (LP relaxation)

---

**Input:** Observation $\boldsymbol{x} \in \mathbb{R}^n$, weight matrix $W = [\boldsymbol{w}_1|\boldsymbol{w}_2|\cdots|\boldsymbol{w}_n]^\top$, initial error bound guess $\epsilon > 0$, scaling factor $\alpha > 1$.
**for** $t = 1, 2, \cdots$ **do**
    Solve the following linear programming:

$$\boldsymbol{z}^{(t)} \leftarrow \arg\max_{\boldsymbol{z}} \sum_i \max\{0, x_i\}\boldsymbol{w}_i^\top \boldsymbol{z}$$

$$\text{s.t } \boldsymbol{w}_i^\top \boldsymbol{z} \leq x_i + \epsilon$$

    $\epsilon \leftarrow \epsilon\alpha$
    **if** $t > 2$ **and** $\exists \boldsymbol{z}^{(t-1)}$ feasible **and** $\|\phi(\boldsymbol{z}^{(t)}) - \boldsymbol{x}^*\|_1 \geq \|\phi(\boldsymbol{z}^{(t-1)}) - \boldsymbol{x}^*\|_1$ **then**
        **return** $\boldsymbol{z}^{(t-1)}$
    **end if**
**end for**

---

We also propose the relaxed LP for LeakyReLU activation, with key step as follows:

$$
\begin{aligned}
\max_{\boldsymbol{z}} \quad & \boldsymbol{x}^\top W \boldsymbol{z} && (9) \\
\text{s.t.} \quad & 1/c\min\{x_i - \epsilon, 0\} \leq \boldsymbol{w}_i^\top \boldsymbol{z} \leq \max\{x_i + \epsilon, 0\}
\end{aligned}
$$

Similarly when $\epsilon = 0$ and $\exists \boldsymbol{z}_0, \text{LeakyReLU}(W\boldsymbol{z}_0) = \boldsymbol{x}$, the solution to (9) is exactly $\boldsymbol{z}_0$.

## B  Theoretical Analysis

### B.1  Hardness

**Warm-up: NP-hardness to Invert a Binary Two-Layer Network:**

As a warm-up, we first present the NP-hardness to recover a binary latent code for a two layer network, and then generalize it to the real-valued case.

**Theorem 5** (NP-hardness to Recover Binary Latent Code For Two-layer ReLU Network). *Given a two-layer ReLU network $G : \{\pm 1\}^k \to \mathbb{R}$ where weights are all fixed, and an observation $x$, the problem to determine whether there exists $\boldsymbol{z} \in \{\pm 1\}^k$ such that $G(\boldsymbol{z}) = x$ is NP-complete.*

To show NP-hardness, we prove that the 3SAT problem is reducible to the exact recovery to the above two-layer network.

Meanwhile, when the ReLU configuration for each layer is given, the recovery problem becomes to solve a simple linear system. Therefore the problem lies in NP, and together we have NP-completeness.

We show that 3SAT is reducible to the inversion problem. We first review the MAX-3SAT problem: Given a 3-CNF formula (i.e. a formula in conjunctive normal form where each clause is limited to at most three literals), determine its satisfiability.

*Proof of Theorem 5.* Now we design a network $G(z) = W_2\text{ReLU}(W_1 z + b_1)$ with binary input vectors that could be reduced from 3SAT problem.

Firstly, let the network consists of $k$ input nodes $z := \{z_1, z_2, \cdots z_k\}$. Next, the connecting layers $u := \{u_1, \cdots u_m\} = \text{ReLU}(W_1 z + b_1)$ consists of $m$ nodes, where each node indicates one clause: $u_i$ will be connected to 3 nodes among $z_j, j \in [k]$, where the weight is $-1$ for a positive literal, and 1 for a negative literal. In other words, $W_1 \in \mathbb{R}^{m \times n}$, where the $i$-th row of $W_1$ is 3-sparse, and corresponds to the 3 variables in the $i$-th clause. Let the bias for each $u_i$ to be $-2$, i.e. $(b_1)i = -2, \forall i \in [m]$. Therefore, only when none of the three literals is satisfied, $u_i$ will output 1, otherwise the ReLU activation will make $u_i$ output 0.

Afterwards, the final layer is one node that takes the summation of $u_i, i \in m$, i.e. $W_2 = \mathbb{1} \in \mathbb{R}^{1 \times m}$. We will set the output to be 0. Therefore only when all $m$ clauses are satisfied, the problem has feasible solutions. Therefore the original problem is also NP-hard.

$\square$

## NP-hardness for Real Network.

*Proof of Theorem 1.* Now we design a real-valued network that could be reduced from 3SAT problem. Firstly, the network consists of $k$ input nodes $z := \{z_1, z_2, \cdots z_k\}$. Next, the connecting 2 layers map each $z_i$ to $v_i = \min\{\max\{z_i, -1\}, 1\}, i \in [k]$. Now, the third connecting layer $u := \{u_1, \cdots u_{m+2}\}$ consists of $m + 2$ nodes, where the first $m$ nodes indicate each clause: $u_i, i \le m$ will be connected to 3 nodes among $z_i, i \in [n]$, where the weight is $-1$ for a positive literal, and 1 for a negative literal. Let $u_{m+1} = \sum_{i=1}^{n} \max\{z_i, 0\}$, and $u_{m+1} = \sum_{i=1}^{n} -\min\{z_i, 0\}$. The bias term on this third layer is $b$ such that the first $m$ values are $-2$ and the last two values are 0. Finally, the last layer $x$ is of 2 nodes, first one is the summation of the first $m$ nodes of $u_i$, and the second one is $u_{m+1} + u_{m+2}$.

We will set the output to be $x = [0, n]$. Notice the first two layers make sure each value of $u_i$ is in the range of $[-1, 1]$. When the output of $x_2 = n$, it means all values of $u_i$ must be $\pm 1$. Therefore we go back to the previous setting with binary input vectors and $x_1 = 0$ simply means that all $m$ clauses are satisfied. Therefore a 4 layered ReLU network could be polynomially reduced from 3SAT problem. $\square$

## Proof of Non-convexity.
The following example demonstrate this property is no longer true for a two-layer case:

**Example 1.** *For $W_1 = [[1, 2], [3, 1]]$, $W_2 = [1, -1]$, and observation $x = 1$, the solution set for*
$$\{z | G(z) = x\}, \text{ where } G(z) \equiv ReLU(W_2 ReLU(W_1 z)),$$
*is non-convex.*

Example 1 is very straightforward to show the non-convexity of the preimage. Notice point $x_1 = (-1, 1)$ and $x_2 = (1, 3)$ are in the solution set, but their convex combination $x_3 = \frac{x_1 + x_2}{2} = (0, 2)$ is not a solution point with $G(x_3) = 2$.

## B.2 Proof of Exact Recovery for the Realizable Case

The proof of Theorem 3 highly depends on the exact inversion for a single layer:

**Lemma 1.** *Under Assumption 1, a mapping $\phi(x) = ReLU(Wx), W \in \mathbb{R}^{n \times k}$ is injective with high probability $1 - \exp(-\Omega(k))$. Namely, when $\phi(x) = \phi(y), x = y$.*

*Proof.* Notice for each $i$-th index, $(Wx)_i$ is positive w.p. $1/2$. Therefore, the number of positive coordinates in $Wx$, denoted by variable $X$, follows Binomial distribution $\text{Bin}(n, p)$, where $n =$

$c_0 k$ and $p = \frac{1}{2}$. With Hoeffding's inequality, $F(k; n, p) := \mathbb{P}(X \leq k) < \exp(-2\frac{(np-k)^2}{n}) = \exp(-\Omega(k))$. Meanwhile, for a matrix with entries following Gaussian distribution, with probability 1 it is invertible. Therefore $\phi^{-1}$ could only have unique solution if there is one. $\square$

Within the proof of Lemma 1, we show that with high probability the observation $\boldsymbol{x} \in \mathbb{R}^n$ has at least $k$ non-zero entries, meaning the original linear programming has at least $k$ equalities. Therefore the corresponding $k$ rows forms an invertible matrix with high probability. Therefore simply by solving the linear equations we will attain the ground truth.

*Proof of Theorem 2.* From Lemma 1, for each layer $\phi_i : \mathbb{R}^{n_{i-1}} \to \mathbb{R}^{n_i}$, with probability $1 - \exp(-\Omega(n_i))$, and for each observed $\boldsymbol{z}_i = \phi_i(\boldsymbol{z}_{i-1}^*)$, by solving a linear system we are able to find $\boldsymbol{z}_{i-1}^*$. By union bound, failure in the whole layerwise inverting process is upper bounded by $\sum_{i=1}^{d} \exp(-\Omega(n_i)) = \exp(-\Omega(k))$, since $n_i > 2n_{i-1}$ for each $i$. $\square$

### B.3 $\ell_\infty$ error bound

With Assumption 2, we are able to show the following theorem that bounds the recovery error.

**Proof of Approximate Recovery with $\ell_\infty$ and $\ell_1$ Error Bound:**
Theorem 3 depends on the layer-wise recovery of the intermediate ground truth vectors. We first present the following lemma for recovering a single layer with Algorithm 1 and then extend the findings to arbitrary depth $d$.

**Lemma 2** (Approximate Inversion of a Noisy Layer with $\ell_\infty$ Error Bound). *Given a noisy observation $\boldsymbol{x} = \phi(\boldsymbol{z}^*) := ReLU(W\boldsymbol{z}^*) + \boldsymbol{e}$. Let $\epsilon = \|\boldsymbol{e}\|_\infty$. If $W$ satisfies Assumption 2 with the integer $m > k$, and the observation $\boldsymbol{z}^*$ has at least $m$ coordinates that is larger than $2\epsilon$, then Algorithm 1 outputs an $\boldsymbol{z}$ that satisfies $\|\boldsymbol{z} - \boldsymbol{z}^*\|_\infty \leq \frac{2\epsilon}{c_\infty}$ with high probability $1 - \exp(-\Omega(k))$.*

*Proof.* Denote $I = \{i | x_i > \epsilon\}$, and $\boldsymbol{x}^* = \text{ReLU}(W\boldsymbol{z}^*)$ to be the true output. Notice it also satisfies $x_i^* > 0, \forall i \in I$ from the error bound assumption. Since $\boldsymbol{x}^*$ has more than $m$ entries $\geq 2\epsilon$, the observation $\boldsymbol{x}$ satisfies $|I| \geq m$. Notice for a feasible vector $\boldsymbol{z}$ with constraints in (6), it satisfies that

$$\|W_{I,:}\boldsymbol{z} - (\boldsymbol{x}^*)_I\|_\infty$$
$$\leq \quad \|W_{I,:}\boldsymbol{z} - \boldsymbol{x}_I\|_\infty + \|\boldsymbol{x}_I - \boldsymbol{x}_I^*\|_\infty \leq 2\epsilon, \tag{10}$$

since the error is bounded uniformly for each coordinate in $\boldsymbol{x}^*$. Meanwhile, notice the real $\boldsymbol{z}^*$ satisfies $\phi_i(\boldsymbol{z}^*) = x_i^*, \forall i \in I$, we have $W_{I,:}\boldsymbol{z}^* = \boldsymbol{x}_I^*$. With Assumption 2, $W_{I,:}$ satisfies $\|W_{I,:}\boldsymbol{a}\|_\infty \geq c_\infty\|\boldsymbol{a}\|_\infty$ for an arbitrary $\boldsymbol{a}$ whp. Therefore together with (10) and let $\boldsymbol{a} = \boldsymbol{z} - \boldsymbol{z}^*$ and get:

$$c_\infty\|\boldsymbol{z} - \boldsymbol{z}^*\|_\infty \leq \|W_I(\boldsymbol{z} - \boldsymbol{z}^*)\|_\infty \leq 2\epsilon. \tag{11}$$

Therefore $\|\boldsymbol{z} - \boldsymbol{z}^*\|_\infty \leq \frac{2\epsilon}{c_\infty}$ with probability $1 - \exp(\Omega(k))$.

$\square$

Theorem 3 is the direct extension to the multi-layer case and we simply apply Lemma 2 from $d$-th layer backwards to the input vector with initial $\ell_\infty$ error of $\epsilon(\frac{2}{c_\infty})^{d-i}$ for the $i$-th layer.

Now we look at some examples that fulfill the assumptions. The proof of $\ell_\infty$ extension is not easy and we look at the following looser result instead.

**Lemma 3** (Related result from [23]). *For a sub-Gaussian random matrix $A$ with height $N$ and width $n$, where $N > 2n$. Its smallest singular value*

$$s_n(A) := \inf_{\|x\|_2=1} \|Ax\|_2.$$

*satisfies $s_n(A) \geq c_2\sqrt{N}$ with high probability $1 - \exp(\Omega(n))$, where $c_2$ is some absolute constant.*

The original paper requires $N > (1+\Omega(\log^{-1}(n))n$ and we presented above with a relaxed condition that $N > 2n$.

*Proof of Corollary 1.* With the aid of Lemma 3, Assumption 2 is satisfied with $m = 2n_{i-1}$ for each layer with high probability. This is because for a random Gaussian matrix $A \in \mathbb{R}^{n \times k}$, $c_2\sqrt{n}\|\boldsymbol{z}\|_\infty \leq c_2\sqrt{n}\|\boldsymbol{z}\|_2 \leq \|A\boldsymbol{z}\|_2 \leq \sqrt{n}\|A\boldsymbol{z}\|_\infty$ w.h.p. Without loss of generality we assume $c_2 \leq 2$. We hereby only need to prove that for each $i$-th layer, $i \in [d]$, the output $\boldsymbol{z}_i^* = \phi_i(\phi_{i-1}(\cdots(\phi_1(\boldsymbol{z}^*))\cdots)) \in \mathbb{R}^{n_i}$

satisfies: $\sum_{j=1}^{n_i} \mathbb{1}_{(z_i^*)_j > \frac{2^{d+1-i}\epsilon}{c_2^{d-i}}} > 2n_{i-1}$ with high probability. We start with the input layer. Notice each entry of $\boldsymbol{y} := W_1 \boldsymbol{z}^*$ follows $\mathcal{N}(0, \sigma_1 = \|\boldsymbol{z}^*\|_2 \sqrt{k})$, $\mathbb{P}(y_j > 2\frac{2^d \epsilon}{c_2^d}) \geq \mathbb{P}(y_j > \frac{\sigma_1}{8}) > 0.45$. Meanwhile, the number of coordinates in $\boldsymbol{y}$ that are larger or equal to $\frac{\sigma_1}{8}$ follows binomial distribution $\mathrm{Bin}(n_1, p), p > 0.45$. Therefore the number of valid coordinates $\geq 0.45n_1 \geq 2k$ (since $n_{i+1} \geq 5n_i, \forall i$) with probability $1 - \exp(-\Omega(k))$. Afterwards since $c_2 < 1/2$ and $\frac{2^{d-i+1}\epsilon}{c_2^{d-i}}, i > 1$ is always smaller than $\frac{\epsilon}{c_2^d}$ and $\|\boldsymbol{z}_i^*\|_2 \geq \|\boldsymbol{z}^*\|_2$ with high probability since the network is expansive, the condition for the remaining layers is easier and also satisfied with probability at least $1 - \exp(-\Omega(n_{i-1}))$. By using union bound over all layers, the proof is complete.

$\square$

The proof for the $\ell_1$ error bound analysis is similar to that of $\ell_\infty$ norm and we only show the essential difference. The key point in transmitting the error from next layer to previous layer is as follows:

$$\|W_{I,:}\boldsymbol{z}_{i-1} - (\boldsymbol{z}_i^*)_I\|_1$$
$$\leq \|W_{I,:}\boldsymbol{z}_{i-1} - (\boldsymbol{z}_i)_I\|_1 + \|(\boldsymbol{z}_i)_I - (\boldsymbol{z}_i^*)_I\|_1$$
$$\leq 2\|(\boldsymbol{z}_i)_I - (\boldsymbol{z}_i^*)_I\|_1$$

(Optimality of Algorithm 2 and $\boldsymbol{z}_{i-1}^*$ being a feasible point)

Together with Assumption 3, we have:

$$\|W_{I,:}\boldsymbol{z}_{i-1} - (\boldsymbol{z}_i^*)_I\|_1 \geq c_1 \|\boldsymbol{z}_{i-1} - \boldsymbol{z}_{i-1}^*\|_1$$
$$\Rightarrow \|\boldsymbol{z}_{i-1} - \boldsymbol{z}_{i-1}^*\|_1 \leq \frac{2}{c_1}\|\boldsymbol{z}_i - \boldsymbol{z}_i^*\|_1.$$

Here $\boldsymbol{z}_i^*$ is the ground truth of $i$-th intermediate vector. $\boldsymbol{z}_i$ is the one we observe and $\boldsymbol{z}_{i-1}$ is the solution Algorithm 2 produces.

## C    More Experimental Results

**More Results on LP Relaxation.**
In Figure 5, we compare the performance with respect to different noise levels over all our proposals, including the results of Algorithm 3 that we omit in the main text. Although we do not see significant improvement of the LP relaxation method over our other proposals, we believe the relaxation over the strict ReLU configurations estimation is of good potential and should be more investigated in the future.

**Time comparison.**
Firstly, we should declare that for the very well-conditioned random weighted networks, gradient descent converges with large stepsize and we don't observe much supriority over GD in terms of the running time. In the table below we presented the running time for random net with different input dimensions ranging from 10 to 110. However, for MNIST dataset, since the weight matrices are not

| k | 10 | 30 | 50 | 70 | 90 | 110 | MNIST(k=20) |
|---|----|----|----|----|----|-----|-------------|
| $\ell_\infty$ LP | 0.63 | 0.73 | 0.83 | 0.90 | 0.95 | 1.03 | 0.5 |
| $\ell_1$ LP | 1.05 | 1.05 | 1.23 | 1.28 | 1.39 | 1.22 | 1.1 |
| LP relaxation | 0.66 | 0.53 | 0.58 | 0.76 | 0.75 | 0.70 | 0.6 |
| GD | 1.59 | 1.65 | 1.72 | 1.80 | 2.09 | 2.01 | 72 |

Table 1: Comparison of CPU time cost averaged from 200 runs, including LP relaxation.

longer well-conditioned, a large learning rate makes GD to diverge, and we have to choose small learning rate 1e-3. The average running time for gradient descent to converge is roughly 1.2 minute, while for $\ell_0$ LP it only takes no more than 0.5 second.

**More experiments on the Sensing Problem.**
Finally we add some more examples for some impainting problem on MNIST with non-identity forward operator $A$.

(a) Uniform Noise; Random Net

(b) Gaussian Noise; Random Net

(c) Uniform Noise; Real Net

(d) Gaussian Noise; Real Net

Figure 5: Comparison of our proposed methods ($\ell_\infty$ LP, $\ell_1$ LP and LP relaxation). As can be shown, all three methods show no significant performance distinction. $\ell_\infty$ LP performs well in most cases except with large Gaussian noise.

Observation

Ground Truth

Ours ($\ell_\infty$ LP)

Gradient Descent [15]

Figure 6: More recovery examples using our algorithm $\ell_\infty$ LP versus gradient descent for an MNIST generative model on some sensing problems.