[Reviews · NeurIPS 2019]

Reviewer 1



Theorem 1 states “The [Recovery of a binary latent code from a two-layer ReLU] is NP-hard since it could be reduced to MAX-3SAT problem.” For an NP-hardness proof, what one wishes to show is a reduction _from_ the known NP-hard problem _to_ the problem being studied. The proof in the appendix appears to correctly use the proposed recovery algorithm as a gadget in solving MAX3SAT, which would be sufficient for an NP-hardness proof, but the text is inconsistent with the theoretical results given. Additionally, while the binary reconstruction problem is interesting for the purpose of the complexity proof, it would be helpful to the reviewers if the rationale for this choice was made clearer outside of the context of the proof and while it is not strictly necessary for the theoretical results it would be useful if it could be shown that the binary latent variable case was representative of the complexity of the problem in general. It is unclear where the expansiveness requirement for the generator model fits into the proof of Theorem 2 The only competing method examined is a strawman version of gradient descent, it would be interesting to see how this method compares to other approaches. Should the inequality in equation 4 be wTz + b <= 0? I’m confused about the dimensions of W and how it relates to both x and z. When defined on line 138, W is nxk. However in assumption 2, W_(I:) is multiplied by x which is nx1. It is my understanding, given that W is nxk, W_(I:) is m(or greater)xk and therefore couldn’t be multiplied by x as in assumption 2. Additionally, on line 125, we see Wz which gives a further indication that W is nxk. Finally, when Wi is defined in theorem 3, it is n_(i-1)xn_i. Therefore, if this was a one layer network, assuming n_0 = k and n_d = n (as stated in line 66) wouldn’t that mean W is kxn? Line 154 states “Let error e satisfies l_inf = epsilon” which seems like a typo Why is epsilon included in algorithm 2 when there’s no longer a maximum value for the error? On line 223 it is mentioned that the uniform distribution and gaussian distribution are in favor of l_0 and l_1 error bound analysis. Isn’t the error bound analysis for l_inf and l1? Wouldn’t gaussian noise imply an l_2 error?

Reviewer 2



In this paper, mainly theoretical in nature, the authors study the invertibility of ReLU (and LeakyReLU) networks, as generative priors for eg denoising or compressive sensing. They recall that a single-layer network can be inverted by a simple Linear Program, and then prove that a two-layers networks is, on the other hand, NP-hard to invert even in the realizable case. They then study the invertibility of networks with random weights one layer at a time, and provide several types of stability bounds in the noisy case, with slightly weaker hypotheses compared to previous results. Finally, they demonstrate that their simple approach can outperform simple gradient descient in some case, for both synthetic and real data, and random and (small) trained network. I enjoyed reading this paper. It is clearly written, and although none of the idea are new here, they are well expressed and some results might be new (the NP hardness, weaker hypotheses to invert random networks). One critic would be that some of the large literature on inverting random networks might be missing, as well as a slightly more thorough comparison with previous results. For instance, the authors prove their result by the classical RIP condition in both l_infty norm (Assumption 2) or l_1 norm (Assumption 3), which has been the case in previous works eg "Towards Understanding the Invertibility of Convolutional Neural Networks" by Gilbert et al., which is not cited. By the way, this is indeed the RIP that is used in both cases, although the authors strangely mention it only in the second case; in compressive sensing the RIP now refers to this property for all general norms (even if the l_infty case is arguably less studied) and has been shown to be equivalent to the existence of a stable decoder in these norms, see eg "Fundamental performance limits for ideal decoders in high-dimensional linear inverse problems" by Bourrier et al. Overall, although the particular results are very probably new, they could be a bt better related to existing works on inverting random neural nets. My other main comment concern the hypothesis that m_i coordinates needs to be bounded away from zero, which is not very natural. Since the matrices are random, it means that the theorem is applicable only for a (random) subset of signals ? If this hypothesis cannot be simplified, it deserved at least to be discussed a bit more: what it the size (whp) of the set of acceptable signals ? If signals are random with a density, is this hypothesis satisfied with high probability ? In the current formulation, it is difficult to know if the theorem is broadly applicable, of assume a shape of signals that is not guaranteed to exist. Minor typos / comments: - l 42: minimima - concerning the NP-hardness, is the non-convex pre-image an entirely separated result, or is it used in the proof ? - l 79-80: the "for any p-norm" is weird, since they are all equivalent - l 101: "in the previous section", I do not believe it was in the previous section ? - l 102 and further: eq (11) is referred numerous times even though it is in the appendix, it is eq (4) instead **EDIT after feedback** I thank the authors for the feedback. I still believe that the "coordinates bounded away from 0" could be further explained, or even suppressed entirely from the statement of the theorems and hidden in the proof, which would be much cleaner in my opinion. I stand by my initial score.

Reviewer 3



In this paper, the authors present an approach for inverting neural network generators. Instead of doing gradient descent, as is more common in the literature, they do a layerwise inversion. In the case where the full output of the network is measured (e.g. denoising), then this method can be directly implemented. In the case where only partial measurements are taken (e.g. compressed sensing, pinpointing, super resolution), then this method can not directly be implemented). The argument relies on the fact that at each layer of the neural network, there are a sufficient number of nonzero activations. This then provides a set of linear measurements, which can be inverted. In the noisy case, the exact linear equations are replaced by a linear program (because there is no direct test for whether the ReLU was active or not. Theorem 3 guarantees approximate recovery if the weights of the network satisfy an L_infinity lower bound, which is satisfied by random weights under a linear amount f expansivity. The approach they study is easy to understand, the analysis of this paper is original and interesting. The theoretical work is reasonably complete, as it contains both the L_infinity and the L1 norm error bounds, which require modifications in the formulation. The scalings the authors obtain are superior to known scalings that work in the gradient descent case, providing theoretical justification to the empirical observation that layerwise inversion can be gradient descent. The empirical studies nicely complement the theoretical analysis. Overall, it is a decent contribution to the field. The paper is mostly clearly written. The last sentence of the abstract can be clarified (better than what?) === Comments after reading rebuttal I was satisfied with the additional experiment the authors reported in the rebuttal. I am upping by rating to an 8 because this additional experiment I think shows that the method is better on real nets than one might originally expect.

[Author Response · NeurIPS 2019]

We thank all the reviewers for their constructive comments and useful suggestions.

**Q (R1): "Comparison with other methods like encoder"** & **"why do we need this technique"**
**A:** This is a very important point that we need to clarify in our paper. GD inversion is not a straw man here: almost all
the prior work on using generative models for solving inverse problems (see reference [3,4,10-14] in our paper) uses
gradient descent as the main inversion technique so improving upon that is significant. There are also methods that
train encoders or even end-to-end reconstruction methods from measurements, but are harder to train and are tuned to a
specific inverse problem as opposed to a general method, see [3]. We will expand on this in the paper.

As compared to GD-based methods, our algorithm is much more efficient. GD costs on average 1.2 minutes to compress
one single image (momentum makes the process faster but still slower than our method). While our algorithm takes
only 0.5 second. See appendix for time comparisons.
We have compared the performance of GD with our method under different network architectures, i.e. different levels
of expansiveness.
Our paper focuses on fundamental theoretical results, since there are very few in this area. In practice our technique can
also be used as a good initialization for gradient-based methods. We verified this for DCGAN. See reply for R3.

**Q (R1): On reductions \*from\* known NP-hard problem A:** You are correct, of course. We fixed the confusing
expression.

**Q (R1): whether the binary case is representative for general case**
**A:** Thank you for your comment. We were able to extend our proof from binary to general real-valued inputs. We
achieve this by constraining an intermediate layer to be binary, using an additional output that enforces an intermediate
layer to have binary values. Combining this with our previous argument establishes that it is NP-hard to invert a 4-layer
network with real-valued inputs.

Specifically, we design a network $f : \mathbb{R}^k \to \mathbb{R}^2$ as follows: After input layer $\mathbf{z} \in \mathbb{R}^k$, we add 2 ReLU layers to make
sure the output of second hidden layer $\mathbf{u} \in [-1, 1]$, i.e. $\mathbf{u} = \min\{\max\{\mathbf{z}, -1\}, 1\}$.[1] Afterwards, we copy the entire
network we used for the binary proof to layer 3 (the original $m$ hidden nodes as layer 3's first $m$ nodes) and to the
output layer $\mathbf{o}$ (the original scalar output as the first observation $o_1$) but add 2 more nodes to layer 3 and one node for
output. The previous binary argument makes sure that if $\mathbf{u}$ has to be binary, we could solve 3SAT. Meanwhile, we let
the two additional nodes on layer 3 to be $a = \sum_i \max\{u_i, 0\}$ and $b = \sum_i \min\{u_i, 0\}$ and the second observation $o_2$
to be $a + b$, which actually satisfies $a + b \equiv \sum_i |u_i|$. At inference time, we let this node $o_2$ to be equal to $k$. In this way
to test for exact recovery, $\sum_{i=1}^k |u_i| = k$ and one has to let each $u_i$ to be +1 or -1.

Therefore we show that for a 4-layer real network, it is NP-hard to determine if it could be exactly recovered for a given
observation.

**Q (R2): Comparison with 'invertibility of convolutional neural networks' or other RIP properties**
**A:** We will add the discussion with (Gilbert et al.) and (Bourrier et al.) as suggested. Our work is substantially different
from (Gilbert et al.) since they still work on linear mappings (convolutional layers without activation functions), while
our work is targeting ReLU or LeakyReLU activations.
Thank you for the pointer of $l_\infty$-RIP property. Typically RIP is for fat matrices with structured input while we deal with
tall matrices. Also we only need the lower bound side of the RIP, so we redefined the condition in the paper. But indeed
the transpose of the weight matrices should satisfy lower bound side of $l_\infty$-RIP. We will add the discussions properly in
the revised version.

**Q (R2): the hypothesis that $m_i$ coordinates need to be bounded away from zero is not natural**
**A:** For the $l_\infty$ case, we have shown that for random matrices, we do not explicitly need this requirement, as shown
in Corollary 1. When the network is expansive and with random weights, there will be enough mass on the positive
observations with high probability.

**Q (R3): Additional Experiments on DCGAN:**

Thank you for your suggestion. We trained a DCGAN architecture using MNIST data. We used 3 convolutional layers
with ReLU activations that are expanding and the last layer being convolutional with sigmoid activation. We used
projected gradient descent (PGD, gradient descent on the last layer, and projection over the first 3 layers) for a denoising
task. We use our algorithm as an initialization for the projection step. Over multiple runs, we compare inversion using
1) GD, 2) GD with momentum, 3) PGD with random initialization (for the projection step), and 4) PGD with our
scheme for initialization. The average relative error for each method was: 1) 0.26, 2) 0.25, 3) 0.17, and 4) 0.088. This
additional experiment shows the benefits of our method for convolutional architectures. We will include more details in
the final paper.

## Footnotes

[1]Here $\max\{\mathbf{z}, -1\}$ could be achieved through $\text{ReLU}(\mathbf{z} + 1) - 1$, and similarly for the $\min$ operation.


[Meta-Review · NeurIPS 2019]

This theoretical paper studied the invertibility of ReLU networks, as generative priors for denoising or compressive sensing. The invertibility of networks with random weights one layer at a time is also investigated and interesting stability bounds are also provided. Note: comments made by Reviewer #2 should be incorporated for the camera ready version.